# Evolution and Optimization of Urban Network Spatial Structure: A Case Study of Financial Enterprise Network in Yangtze River Delta, China

**Yizhen Zhang** [1,2,3]**, Tao Wang** [1,2,3,4,]*****, Agus Supriyadi** [1,2,3] **, Kun Zhang** [5] **and Zhi Tang** [6,7]

1   School of Geography, Nanjing Normal University, Nanjing 210023, China; 191302031@stu.njnu.edu.cn (Y.Z.); 31173018@stu.njnu.edu.cn (A.S.)
2   Key Laboratory of Virtual Geographic Environment (Nanjing Normal University), Ministry of Education, Nanjing 210023, China
3   State Key Laboratory Cultivation Base of Geographical Environment Evolution (Jiangsu Province), Nanjing 210023, China
4   Jiangsu Center for Collaborative Innovation in Geographical Information Resource Development and Application, Nanjing 210023, China
5   College of Tourism, Huaqiao University, Quanzhou 362021, China; zhangkun@stu.hqu.edu.cn
6   School of Urban and Regional Science, East China Normal University, Shanghai 200062, China; tangzhi0923@mail.ahnu.edu.cn
7   Institute for Global Innovation and Development, East China Normal University, Shanghai 200062, China
*   Correspondence: wangtao@njnu.edu.cn

**Abstract:** The urban network is an important method of spatial optimization, and measuring the development level of the urban network is a prerequisite for spatial optimization. Combining geographic information system (GIS) spatial analysis, social network analysis, and multidimensional scaling models, we explored the evolution of the urban network spatial structure in the Yangtze River Delta from 1990 to 2017 and proposed corresponding optimization measures. The results showed that the urban network spatial structure of the Yangtze River Delta has evolved from a single-center cluster with Shanghai as its core to a multi-center network with Shanghai as its core and Nanjing, Hangzhou, and Hefei as secondary cores. The density of the urban network has gradually expanded, but the strength of the connection between edge cities such as Chizhou, Suqian, and Quzhou and the core cities needs to be further improved. We found that the evolution of the urban network spatial structure has been driven by preferential attachment, path dependence, and differences in economic and industrial development. Finally, we propose optimizing the urban network spatial structure by strengthening the driving ability of the core cities, clarifying urban functions and development directions, and establishing a unified coordination mechanism. This paper enriches and deepens our understanding of the characteristics of the city network in the Yangtze River Delta, and provides a reference for the optimization of the urban network spatial structure.

**Keywords:** urban network; spatial optimization; financial enterprise; GIS; Yangtze River Delta; China

---

## 1. Introduction

Since the 1970s, production segmentation, multinational corporations, outsourcing, and innovations in information technology have promoted the evolution of the spatial structure and development of urban networks [1,2]. The development of urban networks has not only accelerated the flow and reorganization of resources, but has also promoted the transition of "space of places" to "space of flows" [3]. As Taylor put it, as a new and highly efficient spatial form, the urban network plays an

important role in regional governance and spatial optimization [4]. Hence, the development of urban networks reflects the evolution of the spatial structure to a certain extent, and measuring the level of urban network development is a prerequisite for spatial optimization.

In 1986, Friedman proposed the concept of a 'world city', which set off a wave of urban network research [5]. With the rapid development of Internet technology and modern transportation, the trend in spatial networking is becoming increasingly obvious, and increasing numbers of theories and practices related to urban networks are being introduced. Some scholars have stated that the networked spatial structure enhances the externalities and spillover effects of economic development and strengthens the synergy and complementarity between cities at different levels [6,7]. Other scholars have reported that the higher the degree a city is embedded in the network, the greater the opportunity for development and the lower the risk of opportunistic behavior [8–10]. With the signing of the United States–Mexico–Canada Agreement (USMCA) and the official implementation of the Horizon 2020 plan, economic and scientific cooperation between different countries and cities have been promoted, thereby further strengthening the development of urban networks [11]. Similar to some countries in North America and Europe, a series of policies to promote regional integration and spatial network development such as the Yangtze River Delta Integration and Guangdong–Hong Kong–Macao Greater Bay Area have been implemented in China [12–14]. These phenomena reflect the importance of strengthening urban network construction as a channel for spatial optimization and high-quality urban development.

Recently, the urban network research based on the corporate network perspective has attracted increased attention from scholars [4,6,11]. As the core of the productive service industry, financial enterprises are important carriers of capital, talent, information, and services, and have gradually become a key driving force for the development of urban networks [9,15]. As revealed by Gomber [16], the study of urban networks based on financial enterprises can more accurately reflect the structure of an urban network. However, due to the difficulty and complexity of obtaining financial enterprise data, most scholars have only studied static urban networks based on data in one time period; research on dynamic urban networks based on multiple time periods is lacking [17,18]. Moreover, most existing studies have used the headquarters of listed financial companies and their subsidiaries to construct urban networks, but this may not accurately reflect the urban networks. Small-and medium-sized enterprises (SMEs) account for more than 99% of the total number of enterprises in the Yangtze River Delta and have become the main force driving economic development [9]. If we focus too much on the urban network formed by listed financial companies, it may lead to too much research focusing on large cities, causing small cities to be neglected [19].

In order to fill the above gaps, this paper provides the following improvements. First, we collected data on the headquarters and branches of all banks, securities, and insurance companies in the Yangtze River Delta region from 1990 to 2017 using the Python programming language. Second, combined with geographic information system (GIS) spatial analysis, social network analysis, and multidimensional scale models, we explored the evolution of the urban network spatial structure in the Yangtze River Delta and proposed corresponding optimization measures. The main advantage of using these data is that financial companies are not only important carriers for the allocation of resources such as capital and technology, but also sensitive to market changes [20]. Moreover, the corporate network formed by the cross-regional layout of financial firms has become a key driving force for shaping urban networks [15]. Finally, two questions need to be answered: What are the characteristics of the evolution of urban network spatial structure and its driving mechanism, and what is the evolution mode of urban networks in the Yangtze River Delta?

The rest of the paper is structured as follows. First, we briefly review the relevant literature in the field of urban networks. Second, we introduce the research areas, data processing, and methods. Third, we combine GIS spatial analysis, a modularity model, and a multidimensional scale model to explore the evolution characteristics of the urban network spatial structure. We also discuss the

factors influencing the urban network and provide some suggestions for the optimization of the urban network spatial structure. Finally, we summarize our conclusions and future research directions.

## 2. Research Background

Spatial structure is the spatial distribution state and spatial combination form of various human economic activities in a specific economic area [21]. It also provides a comprehensive reflection of politics, economy, society, culture, production, and natural conditions [22]. The evolution of regional spatial structure is affected by many factors such as politics, economy, culture, and industry [23]. In the 1930s, the central place theory proposed by the German urban geographer Christaller was considered as one of the basic theories of regional spatial structure and urbanization [24]. Central place theory mainly believes that there is a strict hierarchical system between cities, and the functions and scope of services undertaken by cities of different levels are different. Central place theory introduced the deductive method and rigorous mathematical logic, and pushed the identification of spatial structure from simple description to the discussion of spatial laws and rules [25]. Following its introduction, central place theory has been widely used in regional spatial planning, and has guided the development of regional spatial structure to a certain extent.

Since the 1980s, with the development of transportation and information technology, production segmentation has become an increasingly common phenomenon [26]. Enterprises began to decompose the originally vertically integrated production process into several relatively independent links and deployed them in different cities [27,28]. The segmentation of the value chain has led to profound changes in the urban system, and the urban network based on product segmentation has begun to form. It is generally thought that the urban network is composed of several cities of different sizes that cooperate with each other [29]. These cities are connected through intermediaries such as rapid transportation and enterprises, which are complementary in function and can generate greater external benefits. In the 21st century, the "Globalization and World Cities Research Group (GaWC)" established at Loughborough University in the United Kingdom broke through the original research limitations and used various relational data such as infrastructure and corporate connections, to advance urban network research [30]. In particular, Taylor's Inter-locking Network Model, based on Sassen's Advanced Productive Services (APS) concept and Castells' "space of flows" theory, has realized the transformation from enterprise connections to urban networks [31].

Research on urban networks is maturing. Scholars agree that the urban network is a dynamic process that continuously evolves [32]. Generally, the evolution of urban network spatial structure is mainly divided into four stages: discrete, point-axis diffusion, core-periphery mode, and polycentric network (Table 1) [33]. In the early stage of regional development, economic activities mainly occur within cities, with relatively few connections between cities. With the deepening of the agglomeration and diffusion of resources, the connections between cities become closer [34]. Moreover, relying on the main roads between cities, the regional spatial structure began to shift toward point-axis diffusion [35]. With the continuous development of the economy and society, the density and connection strength of urban networks gradually increase and a multi-directional, multi-dimensional network core skeleton gradually emerges [36]. However, some cities with weaker connections are still scattered on the edge of urban networks. For example, in the European Union's (EU) cooperative research network, Germany, France, and Spain have long occupied the core positions of the network, while Lithuania and Malta are at the edge of the network [11]. With the accumulation, superposition, radiation, and diffusion of various resource elements such as funds and talent, the depth and intensity of city embedded networks have also increased, and the regional spatial structure has begun to evolve into a multi-centric network [37]. At present, most urban agglomerations with a high level of economic development basically exist in the form of a polycentric network. For example, the Atlantic city clusters in the northeastern United States are mainly composed of core cities including Boston, New York, and Washington. The urban agglomerations in northwestern Europe are mainly composed of core cities such as Paris, Amsterdam, and Rotterdam.

**Table 1.** Evolution of the regional spatial structure

| Stage of Development | Discrete | Point-Axis Diffusion | Core-Periphery | Polycentric Network |
|---|---|---|---|---|
| Spatial structure |  |  |  |  |

Urban networks have also been widely used in urban and regional planning, and have gradually become an important tool for urban spatial planning and regional policies [38]. The introduction of the idea of a network into the field of urban planning and spatial policy has created new connotations in spatial planning and policy practice [39]. The form of networked space governance has also become the mainstream idea of regional planning. First, the "multi-center balanced development" and "multi-level governance" advocated by urban networks can effectively improve the efficiency of economic development and promote coordinated regional development. For example, the European Spatial Development Strategy (ESDP) focuses on the construction of urban networks. In the ESDP document, urban networks are described as a compensation for the imbalances and shortcomings of European and transnational urban networks [40]. They state that building a multi-center network spatial structure will improve the city's functions and structure. Notably, competition will become increasingly fierce in the context of the development of "space of flows". Not all cities can benefit from network connections, and those cities that lack competitive advantages will be marginalized [41]. Second, some countries hope that the urban network can enhance the comparative advantages of cities, promote the development of urban specialization and characteristics, and enhance the overall competitiveness of the region. For example, China's integrated development plan of the Yangtze River Delta states that a development pattern with reasonable division of labor and complementary advantages should be formed to promote the overall development of the region [42]. Finally, compared to the central place system, the urban network adopts a path of mutual cooperation and hierarchical development, which tightens the originally loose regional connections. For example, the Dutch Spatial Strategy Plan states that cities of different scales and levels will be connected to enhance the cohesion of cities and build a network economy and a network society. Therefore, the Dutch government decided to prioritize the development of six national-level city networks including Randstad, Brabant, and Limburg South.

In summary, the urban network is an inevitable product of economic development in a certain period and an important manifestation of high-quality and high-level regional development. Under the background of the development of informatization and economic globalization, the urban network has fully penetrated into all aspects of regional development, urban planning, production, and life. The city network is a function of regional development, and the function and effect of the city are more dependent on the urban network.

## 3. Data and Methods

### 3.1. Study Area

The Yangtze River Delta urban agglomeration is located in the lower reaches of the Yangtze River in China (Figure 1). It is an important intersection of the 'Belt and Road' and the 'Yangtze River Economic Belt' and the sixth largest urban agglomeration in the world [43]. The Yangtze River Delta urban agglomeration is composed of 41 cities under the jurisdiction of Shanghai, Jiangsu, Zhejiang, and Anhui Provinces. As of 2019, the Yangtze River Delta region accounted for more than 20% of China's GDP and 3.7% of China's land area. In November 2018, the integrated development of the Yangtze River Delta became a national strategy. The plan stated that it is necessary to enhance the innovation and competitiveness of the Yangtze River Delta region, and increase the degree of economic agglomeration to promote the integrated development of the region. Hence, strengthening

the integrated development of the Yangtze River Delta is essential for optimizing the regional spatial pattern and promoting the high-quality development of China's economic industry.

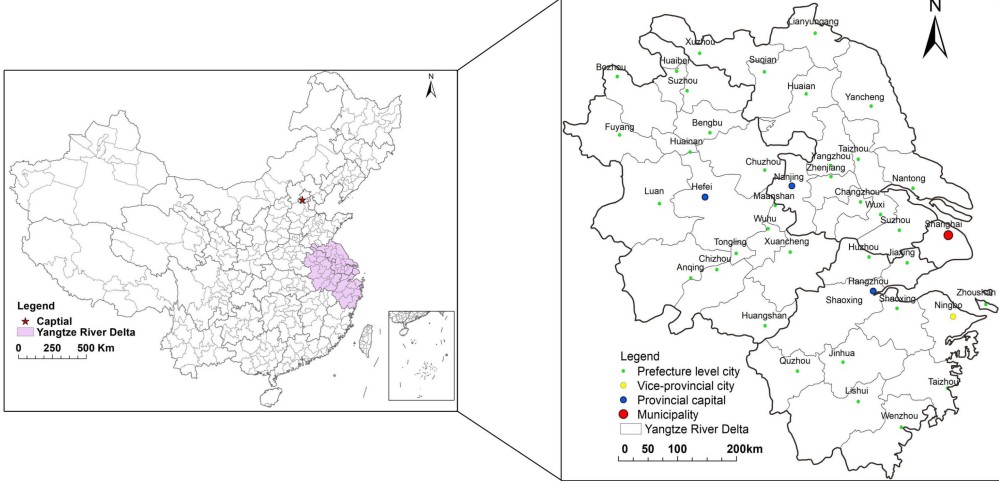

**Figure 1.** Study area and location of the Yangtze River Delta in China.

### 3.2. Data Sources and Network Construction

We mainly used Python programming language to collect the data of the headquarters and branches of securities, banking, and insurance industries in the Yangtze River Delta from 1990 to 2017 [9]. To prevent some financial companies from being missed, or collecting companies that are not financial companies, we set the keywords of the programming language to securities, banking, and insurance and crawled again. The data were filtered and processed according to the following conditions. First, the corporate headquarters had at least one branch in the Yangtze River Delta, and the corporate headquarters and the branch belonged to different cities. Second, information such as the company's registration time, registered capital, and location were queried through the website of the Industry and Commerce Bureau. In 1990, 2000, 2008, and 2017, there were 86, 1807, 4928, and 8051 companies meeting the above conditions, respectively (Figure 2). Finally, referring to the relevant literature [15], we set the weights of associates, joint ventures, local branches, regional branches, and company headquarters to 1, 1, 2, 3, and 4, respectively. A $41 \times 41$ spatial correlation matrix was constructed with cities as the carriers.

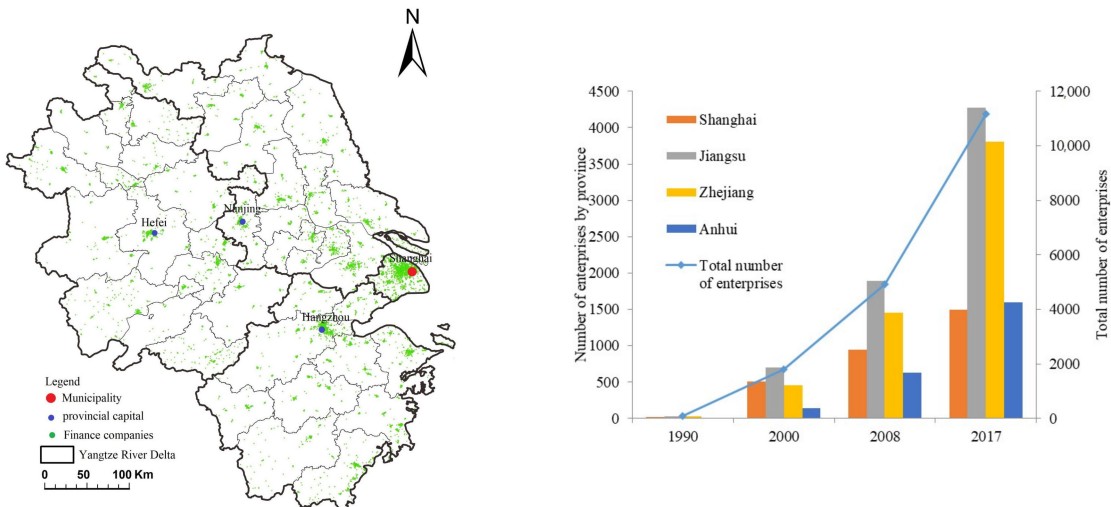

**Figure 2.** Location and number of financial companies in the Yangtze River Delta.

*3.3. Methods*

3.3.1. Interlocking Network Model

The interlocking network model uses enterprises as intermediaries and assigns weights to cities according to the size and level of enterprises [28]. Constructing an urban network through an interlocking network model and calculating its connection strength is mainly divided into two steps. First, suppose that the headquarters of enterprise $k$ is located in city $i$, and its branch (regional branches, local branches, associates, and joint ventures) is located in city $j$. Then, the connection strength between city $i$ and city $j$ based on enterprise $k$ is:

$$R_{ij,k} = 4(3C_{rb} + 2C_{lb} + C_{as} + C_{ve})$$

(1)

where $R_{ij,k}$ represents the connection strength between cities $i$ and $j$ based on $k$ enterprises. There may be $m$ such enterprises $k$ whose headquarters are located in city $i$ and their branches (regional branches, local branches, associates, and joint ventures) are located in city $j$. Therefore, the total connection strength between city $i$ and $j$ is:

$$R_{ij} = \sum_{k=1}^{m} 4(3C_{rb} + 2C_{lb} + C_{as} + C_{ve})$$

(2)

where $R_{ij}$ represents the strength of connections between cities $i$ and $j$ based on $k$ companies, and $m$ represents the number of financial company headquarters. $C_{rb}$, $C_{lb}$, $C_{as}$, and $C_{ve}$ respectively represent the number of enterprise $k$ whose headquarters are located in city $i$, but its regional branches, local branches, associates and joint ventures are located in city $j$. Coefficient *4* represents the weight of the headquarters, and coefficients *3*, *2*, *1*, and *1* represent the weight of branches (regional branches, local branches, associates, and joint ventures, respectively).

3.3.2. Degree Centrality

Degree centrality reflects the connectivity of the urban network. Generally, the higher the degree centrality, the stronger the connectivity of the city [44]. The formula is as follows:

$$DC_i = \sum_{a=1}^{n} R_{ij}(i \neq a)$$

(3)

where $R_{ij}$ is the connection strength between cities $i$ and $j$ ($i \neq a$); $DC_i$ represents the total connection strength between city $i$ and other cities in the network; and n represents the number of cities in the area.

3.3.3. Network Density

Network density refers to the closeness of connections between cities. The greater the network density, the better the overall connectivity of the urban network [45]. Its formula is:

$$D = \sum_{i=1}^{n} \sum_{j=1}^{n} d_i(c_i, c_j) / n(n-1), (i \neq j)$$

(4)

where $D$ represents the urban network density; $n$ represents the number of node cities; and $d_i(c_i, c_j)$ is the amount of connection between nodes $c_i$ and $c_j$. The greater the $D$, the greater the density of the urban network and the closer the connections between cities.

### 3.3.4. Modularity

Modularity is an optimization algorithm based on multi-level spatial networks. It can be used to quickly and accurately discover the community and describe the intimacy of the community, and is one of the best community discovery algorithms [46]. The degree of modularity is defined as the ratio of the total number of edges in the community to the total number of edges in the network minus the expected value. The formula for calculating the modularity ($Q$) is:

$$Q = \frac{1}{2m} \sum_{ij} \left[ A_{ij} - \frac{k_i k_j}{2m} \right] \delta(c_i, c_j) \tag{5}$$

where $A_{vw}$ is the weight between nodes $v$ and $w$; $k_v$ and $k_w$ refer to the degree values of nodes $v$ and $w$, respectively; and $\delta(c_v, c_w)$ is used to judge whether nodes $v$ and $w$ are divided into the same community. If so, a value of *1* is assigned; otherwise it is *0*. The network is usually divided into n communities, and there is an n-dimensional matrix. To simplify the calculation, the calculation of $Q$ is changed as follows [47]:

$$
\begin{aligned}
Q &= \frac{1}{2m} \sum_{ij} \left[ A_{ij} - \frac{k_i k_j}{2m} \right] \sum_v \delta(c_i, v)(c_j, v) \\
&= \sum_v \left[ \frac{1}{2m} \sum_{ij} A_{ij} \delta(c_i, v) \delta(c_j, v) \right] - \frac{1}{2m} \sum_i k_i \delta(c_i, v) \frac{1}{2m} \sum_j k_j \delta(c_j, v) \\
&= \sum_v \left( e_{vv} - a_v^2 \right)
\end{aligned}
\tag{6}
$$

$$e_{vw} = \frac{1}{2m} \sum_{ij} A_{ij} \delta(c_i, v)(c_i, w) \quad a_v = \frac{1}{2m} \sum_{ij} k_i \delta(c_i, v) \tag{7}$$

where $e_{vw}$ is the ratio of the sum of the number of edges in community v and community $w$ to the total number of edges, and $a_v$ represents the ratio of the number of all edges in community $i$ to the total number of edges. The principle of operation is shown in Figure 3:

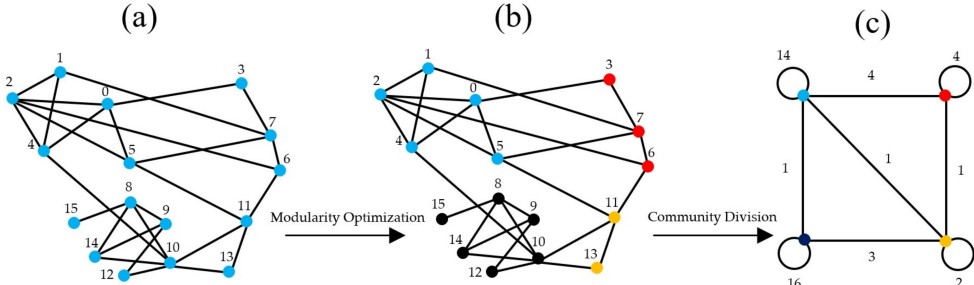

**Figure 3.** Operation principle of the modularity model. We divided closely connected cities into different communities through a modularity model. (**a**) The blue nodes represent all the different cities. (**b**) After modularity optimization, some closely connected cities are selected, and cities of the same color are in the same community. (**c**) The cities of the same color are grouped into the same community after modularity division.

### 3.3.5. Multidimensional Scaling

Multidimensional scaling (MDS) is used to construct a proximity relationship matrix based on the similarity between research objects to analyze the degree of fit through a stress coefficient (Stress), and reduces dimensionality through clustering or dimensional analysis to clarify the spatial structure clearer [48]. Although the MDS model is rarely used in the field of geospatial networks, this method is conducive to simplifying and visualizing spatial problems to better understand spatial relationships. We used MDS to fit the urban network spatial structure of the Yangtze River Delta in the space-time dimension. The calculation formula is:

$$S = \left[ \sum_{i=1}^{n} \sum_{j=1}^{n} (d_{ij} - \hat{d}_{ij})^2 \middle/ \sum_{i=1}^{n} \sum_{j=1}^{n} (d_{ij} - \bar{d})^2 \right]^{1/2} \tag{8}$$

where $d_{ij}$ is the original distance of the research object; $\hat{d}_{ij}$ is the predicted distance of the research object under the spatial representation; and $\bar{d}$ is the average of the original distance of all the research objects. In this paper, the operation steps of the *MDS* model are as follows:

- The strength of connections between cities in *1990*, *2000*, *2008*, and *2017* is processed into a matrix form that can be recognized by *SPSS* software.
- The processed data are imported into *SPSS* software, and the Stress coefficient in the *MDS* model is used to analyze its fitting degree. Generally, a Stress coefficient less than 0.2 indicates a better fitting effect.
- *SPSS* software is used to generate a spatial perception map that represents the relationship between cities based on the strength of connections between them. Through the spatial perception map combined with previous research, we can abstract expression of its spatial relationship.

3.3.6. Quadratic Assignment Procedure

The quadratic assignment program (*QAP*) compares the similarity between the lattice values of one matrix and multiple matrices based on the permutation of relational matrices, and provides the correlation coefficient and regression coefficient. The difference between the *QAP* model and other standard statistical procedures is that the values of the matrix are not independent of each other [49]. Hence, many standard statistical procedures cannot perform parameter estimation and statistical testing, as the standard deviation will be incorrectly calculated. The *QAP* model also circumvents the problem of multicollinearity in the traditional multiple regression model, effectively reducing the impact of noise. The steps for using the *QAP* model in this study were as follows:

- We constructed a $41 \times 41$ spatial matrix of the connections between cities (the $R$ value calculated above) and used it as the dependent variable ($Y$).
- With reference to relevant literature, we selected some factors that may affect the urban network, and constructed these factors into a $41 \times 41$ spatial matrix.
- We took the city connection matrix as the dependent variable ($Y$) and the influencing factor matrix as the independent variable ($X$), and imported them into the *QAP* relational regression model for regression analysis. This step was implemented in UCINET.

## 4. Result

*4.1. Spatial Connection Identification*

The intra-firm network approach simulates many more actual inter-city linkages [50]. We visualized the evolution of an urban network spatial structure based on ArcGIS10.2 (Figure 4). Overall, the strength of the urban network connections continued to increase over time, resulting in increasing urban network density. The evolution of inter-city linkages was driven by network mechanisms such as hierarchical diffusion, preferential attachment, and path dependence. The spatial distribution of urban connections was unbalanced and had strong network hierarchical characteristics.

In the early stages of urban network development, to obtain more externalities, cities were more inclined toward proximity and directional linkages [7]. For example, in 1990, Shanghai's connection strength with Suzhou, Jiaxing, and other surrounding cities was significantly higher than other cities. On one hand, various forms of proximity including geographic and cultural proximity tend to reshape the urban network structure. On the other hand, the prerequisite for the development of urban networks is perfect road–infrastructure networks. However, road–infrastructure networks in the Yangtze River Delta were poorly developed before 1990, which restricted the cross-regional flow of

resources and hindered the development of urban networks. Since 2000, Shanghai has gradually broken through administrative barriers and spatial boundaries, and has begun to connect with network edge cities such as Fuyang and Quzhou, but the connection strength remained relatively weak.

In 2008, as the cost of urban network connections continued to decline and the pattern of urban network connections changed dramatically, inter-provincial links between Nanjing–Hangzhou (112) and Hangzhou–Hefei (48) have gradually appeared, and the strength of connections has increased. The network connection displayed a certain path dependence effect due to the embeddedness of the urban network. For example, a lock-in effect was identified in the inter-city connections in the Yangtze River Delta that was mainly concentrated between Shanghai–Hangzhou, Shanghai–Nanjing, Shanghai–Suzhou, Shanghai–Wuxi. and Shanghai–Hefei. During the study period, the strength of the connections between the five fixed cities (between Shanghai and Nanjing, Suzhou, Wuxi, Hangzhou, and Hefei) accounted for more than 20% of all connections. From 2000 to 2017, Shanghai–Hangzhou, Shanghai–Nanjing, Shanghai–Suzhou, and Shanghai–Wuxi all ranked among the top five in terms of connection strength. This indicated that the connections among fixed city–dyads maintained strong self-reinforcement. In other words, the dynamic evolution of inter-city connections maintained notable spatiotemporal inertia. In 2017, the inter-provincial connections gradually increased, and the strength of inter-city connections increased significantly when compared with 2008. For example, the connection strengths of Nanjing–Hangzhou and Hangzhou–Hefei were 280 and 196, respectively, which were much higher than the 112 and 48 in 2008. This is not surprising as the Chinese government believes that the integrated development of the Yangtze River Delta is important for shifting the Chinese economy from high speed to high quality, and is also a key measure for promoting coordinated regional development. Since 2009, relevant policies have been implemented to support the integrated development of the Yangtze River Delta.

However, the network edge cities such as Chizhou, Suqian, and Quzhou had relatively weak connections with core cities. The spatial distribution of urban connections was unbalanced and showed strong network hierarchical characteristics. This is not conducive to the integration and coordinated development of the Yangtze River Delta. Moreover, related research shows that road–infrastructure networks are the prerequisite for resource circulation. Any country or city that wants to strengthen economic and industrial development and promote inter-city links must first strengthen the construction of road–infrastructure networks. In the future, it is necessary to optimize the spatial distribution of resources, strengthen the construction of road–infrastructure networks between edge cities and core cities, and improve the network status of edge cities.

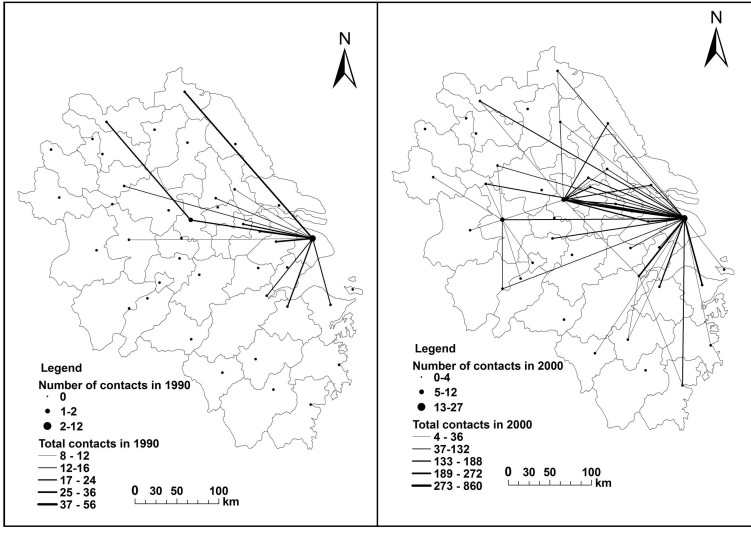

**Figure 4.** *Cont.*

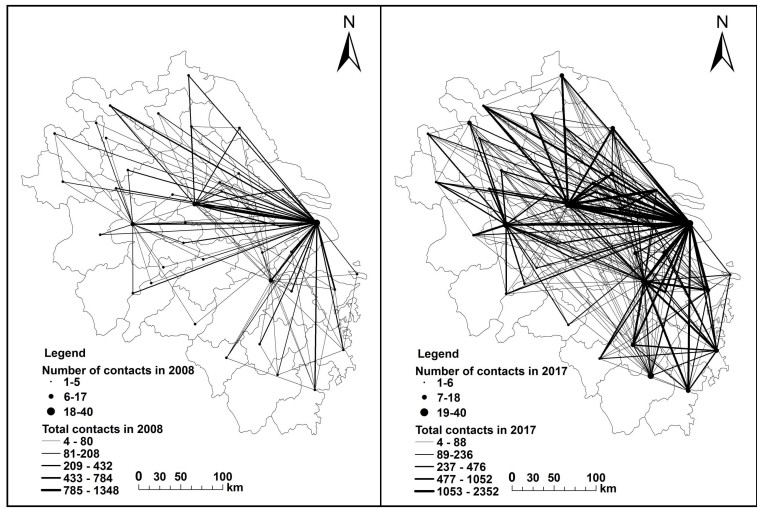

**Figure 4.** Intercity network for the Yangtze River Delta from 1990 to 2017. The weight of the edge represents the R value (the total connection strength between cities) calculated above.

*4.2. Community Division of Urban Network Spatial Structure*

Recently, various forms of planning including the Yangtze River Delta regional integration development plan, Shanghai metropolitan area, and Nanjing metropolitan area have promoted the spatial networking development of the Yangtze River Delta. However, due to the effects of preferential attachment and spatial proximity, the connections between cities often show path dependence, which is mainly manifested in the generation of different network communities [51,52]. Therefore, to understand the spatial structure of the urban network more intuitively, the R value calculated in the previous section was used as the weight to divide the network community through the modularity model (Figure 5).

In 1990, two network communities formed with Shanghai and Nanjing as the core. The Shanghai network community included 11 cities such as Hangzhou and Hefei, while the Nanjing network community only included Suzhou. This may have occurred due to the socialist system with Chinese characteristics, so most of the high-quality political and economic resources were mainly concentrated in Shanghai, resulting in more connections between Shanghai and other cities. Since 2000, the characteristics of the network community with Shanghai and Nanjing as the core have further strengthened. The members of the Shanghai network community mainly include Hangzhou, Suzhou, Hefei, and another 14 cities, which is an increase of three compared with 1990. The members of the Nanjing network community mainly included five cities in Jiangsu Province: Yangzhou, Yancheng, Wuxi, Zhenjiang, and Changzhou, which showed strong geographical proximity effects. However, the provincial capitals Hangzhou and Hefei did not form independent network communities. The main reason for this finding is that Shanghai and Hangzhou, as the core cities of the Yangtze River Delta, had similar economic development and industrial structure, and close cultural and technological exchanges. However, the situation in Hefei was different. In 2000, the economic development level of Anhui Province was relatively low, and the connections between cities were weak. Hefei had not yet played a key role in promoting the development of other cities in Anhui Province, and only had established links with Shanghai, which was highly economically complementary. In 2008, the spatial structure of the network community changed; meanwhile, three network communities with Shanghai, Nanjing, and Hefei as the core were formed. The cities in the Shanghai network community were mainly from Zhejiang Province, the cities in the Nanjing network community were all from Jiangsu Province, and the cities in the Hefei network community were all from Anhui Province. The administrative divisions and geographic distance were important factors that affected the division of network communities. It is worth noting that Hangzhou, the capital of Zhejiang Province, had not yet formed an independent network community, but had become an important part of Shanghai's network community with

other cities in Zhejiang Province. In 2017, with the development of transportation infrastructure and the improvement of the institutional environment, the urban network spatial structure was further optimized. Four network communities with Shanghai, Nanjing, Hangzhou, and Hefei as the core formally formed.

The latest document of the State Council of China on the integration of the Yangtze River Delta reported that the Yangtze River Delta should take Shanghai as the core to drive the overall development of the region, and Nanjing, Hangzhou, and Hefei as the sub-cores to drive the coordinated development of Jiangsu, Zhejiang, and Anhui Provinces, respectively [42]. Based on the actual situation, the current urban network pattern in the Yangtze River Delta is also consistent with the network community proposed in this paper. This also shows that the multi-center network community with Shanghai as the core and Nanjing, Hangzhou, and Hefei as the secondary cores is in line with the current urban development interests and is a relatively ideal urban network spatial structure.

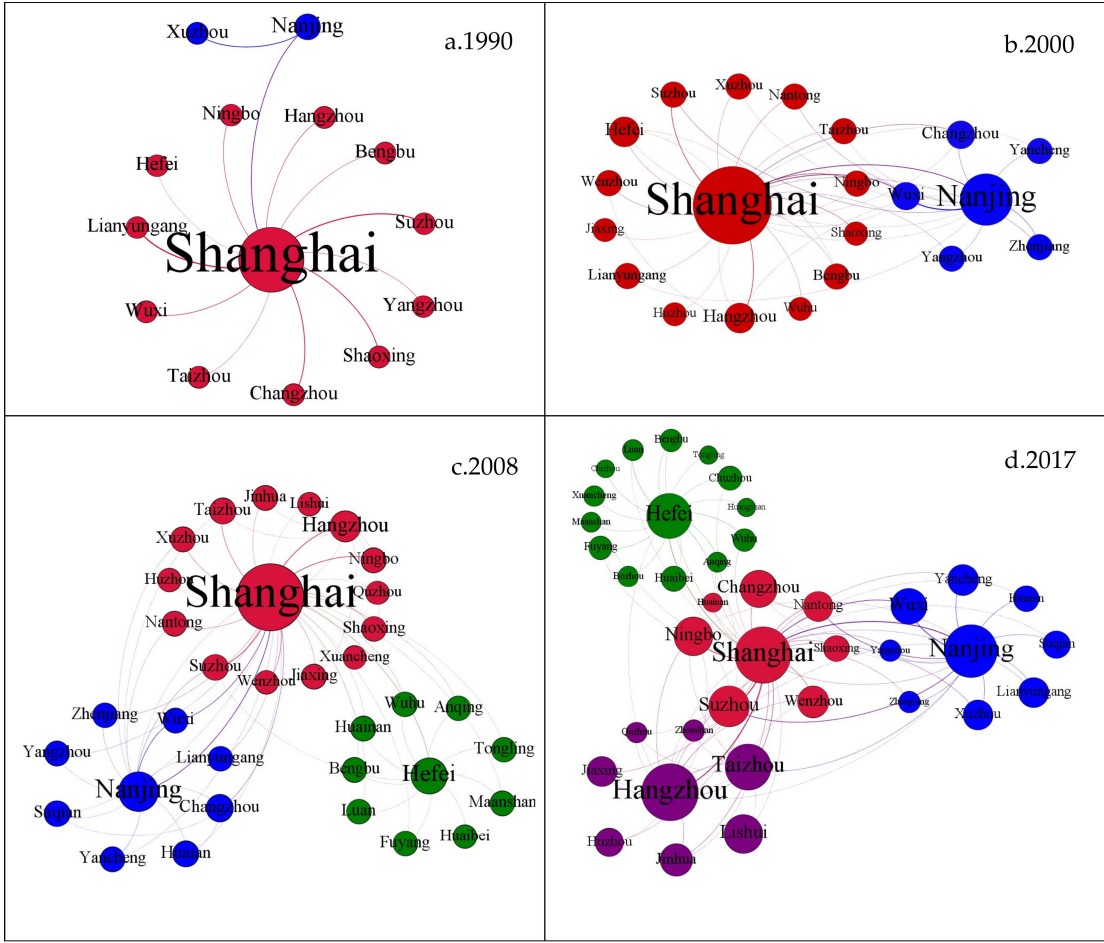

**Figure 5.** Network community division in the Yangtze River Delta from 1990 to 2017. The weight of the edge represents the R value (the total connection strength between cities) as calculated above. The same color is used to represent the cities in the same community, and the size of the node represents the degree centrality. (**a**–**d**) show the evolution of network community in 1990, 2000, 2008, and 2017, respectively.

### 4.3. Evolution Mode of the Urban Network Spatial Structure

Based on the urban connection strength matrix in different years, the MDS model was used to generate the spatial perception map shown in Figure 6. For 1990, 2000, 2008 to 2017, the Stress coefficients were 0.09, 0.12, 0.08, and 0.06, respectively, and the fitting effect was good. Figure 6 depicts the changes in the relative positions and interrelationships of cities in the Yangtze River Delta from 1990 to 2017, which allowed us to further judge the development process and evolution mode of the

spatial structure of the Yangtze River Delta. Overall, the urban network of the Yangtze River Delta has evolved from a single-center cluster centered on Shanghai to a multi-center network model centered on Shanghai, Nanjing, Hangzhou, and Hefei.

### 4.3.1. Single-Center Agglomeration Mode

In 1990, Shanghai was overwhelmingly dominant within the Yangtze River Delta region with its unique political and economic resources. Due to the limitation of geographical location and transportation facilities, Shanghai's external radiation capacity was weak, showing a single-center polarization development mode (Figure 6a). This has led to the development gap between Shanghai and neighboring cities, and further aggravated the uneven economic development of the Yangtze River Delta. Hence, some cities far away from Shanghai and cities with lower economic and industrial development levels drifted on the periphery of the urban network.

### 4.3.2. Point-Axis Diffusion Mode

In 2000, with the further decline in the cost of intercity connection, and the rapid development of road–infrastructure networks, core cities such as Shanghai, Hangzhou, and Nanjing rely on the Shanghai–Hangzhou and Shanghai–Nanjing expressways to gradually increase the strength of connections between cities. The strength of connections between core cities such as Shanghai, Hangzhou, and Nanjing and surrounding cities have especially increased over time (Figure 6b). However, the connections between other cities remained weak. First, this indicated that the dynamic evolution of urban networks maintained notable spatiotemporal inertia. Hence, second-tier cities have more opportunities to absorb resources and economy from the top cities. Moreover, compared with small cities, second-tier cities have better economic and industrial absorption capabilities. This is not surprising as the urban network is a manifestation of the city's deep participation in the division of labor and collaboration with its advantages in resources, capital, and talent. Generally, the larger the city, the richer its resources and the higher its network status. Second, it showed that the connections between cities were generally spread along main traffic lines. In other words, urban road–infrastructure networks determine the development of urban networks to a certain extent.

### 4.3.3. Core-Periphery Mode

In 2008, as the inter-city connection was further strengthened, the second-tier cities around the core city gradually developed. The network connection range and radiation-driving capabilities of core cities and second-tier cities gradually increased, resulting in a continuous increase in network density. However, due to the constraints of natural conditions, economic development level, and transportation facilities, some cities did not fully participate in the connection of the urban network and were still on the periphery of the urban network (Figure 6c). For example, Suqian, Chizhou, and Fuyang have fewer connections with other cities and lower network status. Meanwhile, the urban network spatial structure gradually changed from a point-axis diffusion model to the core-periphery mode.

### 4.3.4. Multi-Center Network Mode

In 2017, the flow frequency of various elements in the region accelerated, and the space-time compression effect gradually appeared, leading to the continuous deepening of regional integration. Moreover, Shanghai's network functions also underwent major changes. In the earlier periods, Shanghai continuously absorbed various resources from the surrounding cities due to its political and economic advantages. In the later periods, Shanghai helped spread technology and develop the economy of the surrounding cities. The transfer of economy and technology is an important means to promote the integration of the Yangtze River Delta and alleviate the pressure on resources and population in large cities such as Hangzhou, Shanghai, and Nanjing. Under the guidance of regional core cities and sub-central cities, the strength of inter-city connections further increased, and the urban network spatial structure began to evolve into a polycentric network mode (Figure 6d).

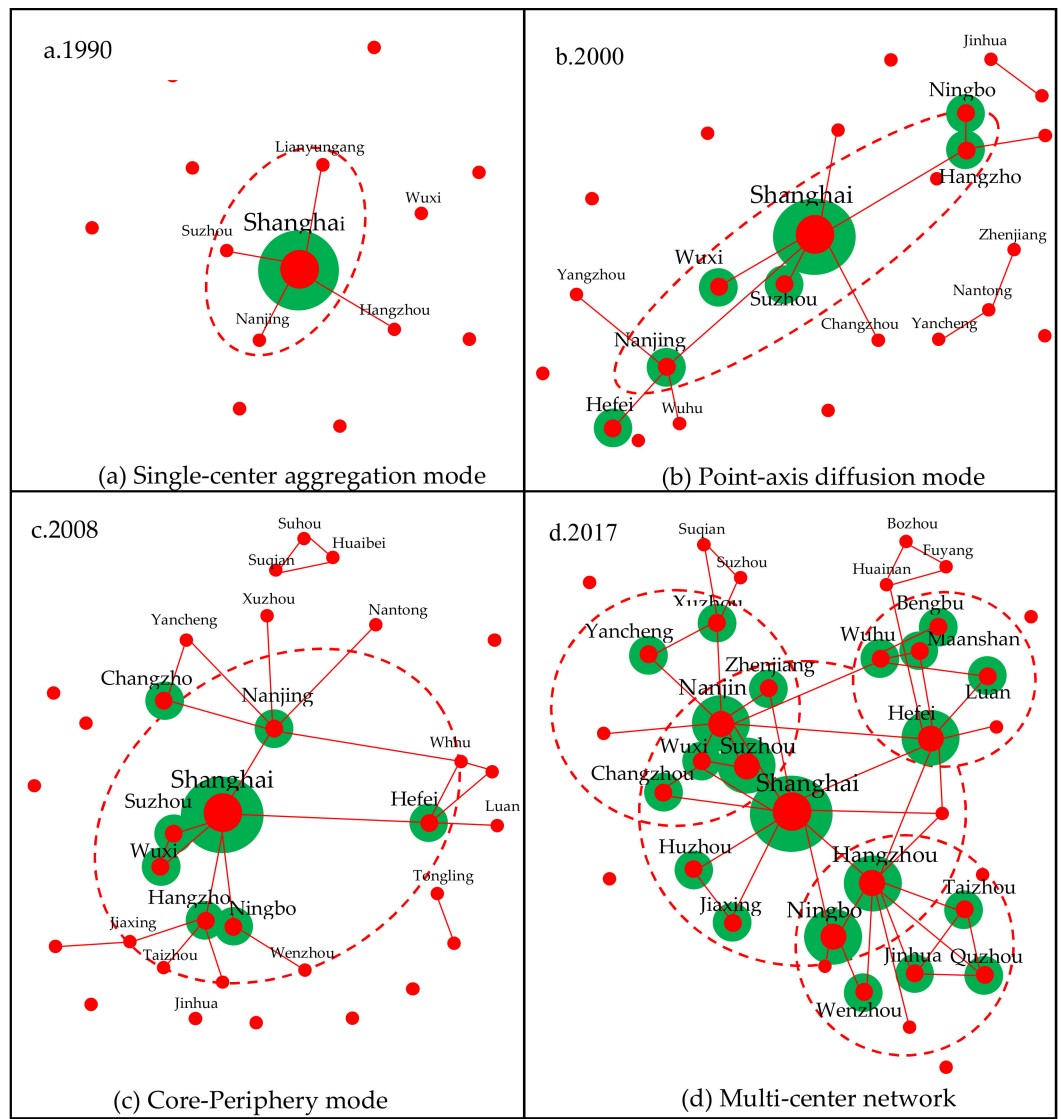

**Figure 6.** Evolution Mode of Urban Network Spatial Structure in Yangtze River Delta from 1990 to 2017. **Note:** 🔴 represents a type I city, 🔴 represents a type II city, • represents a type III city, • represents a type IV city. The green surrounding the red node refers to the research of Li et al., and to a certain extent represents the urban hinterland. [14].

### 4.4. Potential Determinants of the Strength of Urban Network Connections

#### 4.4.1. Index Selection and Relationship Regression Model Construction

In essence, the urban network is actually a relational network. This means that many standard statistical methods including ordinary least squares (OLS) are struggling to remove the influence of relationships to evaluate them. However, the QAP model uses matrix lattice value comparison to explain relationships based on relationships. Therefore, the root cause of the correlation of independent variables is solved [49,53]. The model is as follows:

$$Y_i = a_0 + a_1x_1 + a_2x_2 + \cdots + a_7x_7$$

where $Y_i$ represents the dependent variable of the urban network relation matrix; $a_0$ is a constant, $a_1$–$a_7$ are regression coefficients; and $X_1$–$X_7$ are explanatory factor relation matrices. According to relevant references and the situation in this study, the factors affecting the urban network were divided

into four levels [15]: differences in economic development, differences in financial development, differences in institutional development, and geographical proximity. Moreover, the difference in economic development is represented by GDP per capita ($X_1$), square of GDP per capita ($X_2$), tertiary industry output value ($X_3$), and fixed asset investment ($X_4$). We selected the variable of the square of the per capita GDP difference to estimate the impact of the further expansion of per capita GDP gap on the urban network. The difference in financial development is represented by a financial-related ratio ($X_5$). The differences in institutional development mainly take into account the behavior of local governments and are expressed in local government fiscal expenditures ($X_6$). Geographical proximity mainly refers to whether two cities are in the same administrative division. We set the geographic proximity value of two cities within the same administrative division to 1, and to 0 in the opposite situation, and then constructed their relationship matrix ($X_7$).

### 4.4.2. Potential Determinants of Urban Networks

With the help of the QAP model in UCINET, a random replacement number of 10,000 was selected for regression analysis. Overall, the results of the correlation analysis and regression analysis were in line with the theoretical analysis and the reality of the Yangtze River Delta and have strong explanatory power (Table 2). The results were as follows: (1) Per capita GDP, the square of per capita GDP, and the urban network had an inverted U relationship, and both passed the significance test. This shows that the relationship between the per capita GDP gap and the urban network could be divided into two stages. When the per capita GDP gap between cities was small, the urban network connection became closer. However, when the per capita GDP gap between cities was large, it hindered the development of urban networks. (2) The regression coefficient of the tertiary industry output value was 0.201, and the significance probability was 0.037,showing that the industrial structure was positively related to the urban network. Specifically, cities with similar industrial structures have more technical and economic connections, which strongly promote urban network development. (3) The regression coefficient of fixed investment was −0.091, which passed the significance test, reflecting that the greater the gap between fixed investments, the smaller the strength of the connection between cities. (4) The regression coefficient of the financial-related ratio was 0.026, and its significance probability was 0.051, showing that although the financial-related ratio was positively related to the urban network, the explanation was weak. (5) The regression coefficient of local fiscal expenditure was −0.078, which passed the significance test. This shows that the greater the gap between local fiscal expenditures, the fewer the connections between cities. (6) The regression coefficient of geographical proximity was −0.025, which failed the significance test. This may be that financial resources have strong liquidity, which makes it difficult for administrative divisions to restrict their flow in geographical space.

**Table 2.** Regression results of urban network influencing factors in the Yangtze River Delta.

| Explanatory Variables | Correlation/Regression Coefficient | | | |
| --- | --- | --- | --- | --- |
| | QAP Related Analysis | | QAP Regression Analysis | |
| | Correlation Coefficient | P | Regression Coefficients | P |
| Differences in economic development | | | | |
| Per capita GDP | −0.123 * | 0.029 | 0.015 ** | 0.033 |
| Square of GDP per capita | −0.215 * | 0.055 | −0.012 ** | 0.037 |
| Tertiary industry output value | 0.210 ** | 0.017 | 0.201 ** | 0.048 |
| Fixed asset investment | −0.046 ** | 0.374 | −0.091 ** | 0.026 |
| Differences in financial development | | | | |
| Financial-related ratios | 0.182 ** | 0.029 | 0.026 * | 0.051 |
| Differences in institutional development | | | | |
| Local government fiscal expenditures | 0.062 * | 0.370 | −0.078 * | 0.091 |
| Geographical proximity | | | | |
| Administrative relations | −0.049 ** | 0.014 | −0.025 | 0.337 |

**Note**: ***, **, and * represent significance at the levels of 1%, 5%, and 10%, respectively.

## 5. Urban Network Spatial Structure Optimization Measures

Based on the research results of this paper, some suggestions for optimizing the urban network spatial structure are proposed.

(1) Strengthen the resource agglomeration and diffusion capabilities of core cities in the network, and facilitate the liberalization of resource flow. The development of spatial networking has enhanced the cities' resource exchange capacities and allocation efficiencies, and reduced the risk of opportunistic behavior. The network core cities have a strong ability to gather and spread resources, and act as a bridge and intermediary in the development of the urban network. However, the high-quality integrated development of the Yangtze River Delta cannot only rely on network core cities. In the future, it is necessary to further strengthen the construction of transportation infrastructure, reduce the cost of factor flow, and improve the connection efficiency of edge cities.

(2) Optimize the spatial distribution of resources and guide the rational positioning of cities. Our research found that network core cities had higher network status and stronger control over network edge cities. It is not surprising that due to the socialist system with Chinese characteristics, a large number of high-quality resources such as policies and funds are mainly concentrated in municipalities and provincial capitals (Beijing, Shanghai, Guangzhou, and Shenzhen). The continuous growth of industries and populations in large cities is creating increasing amounts of economic and social conflict. However, some small cities suffer from population loss and lagging economic and industrial development due to the lack of resources. Therefore, we must change the past development concept, optimize the spatial allocation of resources, and promote the coordinated development of urban population–economy–industry. It should also be noted that coordinated development is not equalized development without difference between cities. It is necessary to clarify the positioning of urban functions, reduce the "siphon effect" of core cities through function transfer and optimizing resource allocation, and guide the specialization and characteristic development of non-core cities.

(3) Through network spillover effects, strengthen the connection strength between network edge cities and core cities. Compared with the traditional spatial structure, the urban network spatial structure can accelerate the flow of resources to take advantage of the spillover effects of the economy, technology, and knowledge of the core cities. The spillover of economy, technology, and knowledge can promote industrial upgrades and technological progress in lower-level cities [54]. However, we also found that network spillover effects are related to factors such as geographic distance and cultural differences. This means that it is difficult for cities at the edge of the network to receive network spillover effects. Therefore, it is necessary to enhance the connection between network edge cities and core cities through industrial transfer and technical guidance.

(4) Construct a multi-center network development mode to promote the integrated development of the Yangtze River Delta. More and more scholars believe that with the continuous development of China's economy, a single-center cluster with Shanghai as the center is not conducive to the coordinated development of the Yangtze River Delta. Our research also shows that the current urban spatial structure of the Yangtze River Delta has developed from a single-center agglomeration to a multi-center network, which may have been driven by a variety of factors such as the market, government, and economy, and is a relatively ideal spatial development mode. However, due to historical factors, cultural differences and policy factors, forming a reasonable division of labor and coordination mechanism among cities in the Yangtze River Delta will be difficult. It may cause vicious competition in economic and industrial development, which is not conducive to the coordinated development of urban networks [55]. Therefore, we must make full use of tangible and intangible approaches to strengthen the docking of urban industries and the division of labor to avoid redundant construction and vicious competition.

## 6. Conclusions

The urban network is an important method of spatial optimization, and measuring the development level of the urban network is a prerequisite for spatial optimization [24]. In this study, we first used the Python programming language to crawl the data of financial companies of different sizes and levels in the Yangtze River Delta in 1990, 2000, 2010, and 2017. Second, combined with the modularity and MDS models, we explored the network community division and network development model, respectively. Finally, we explored the factors influencing the urban network and provided some suggestions for the optimization of the urban network spatial structure. Our findings enrich and deepen the understanding of the urban network characteristics of the Yangtze River Delta, and provide a reference for optimizing urban network spatial structure.

Consistent with other research, we found that different forms of proximity including cultural and geographic proximity tend to shape the urban network structure [6,40,42]. We also found that the network connection has a certain path dependence effect due to the embeddedness of the urban network. For example, a lock-in effect was found in the inter-city connections in the Yangtze River Delta, which was mainly concentrated between Shanghai–Hangzhou, Shanghai–Nanjing, Shanghai–Suzhou, and Shanghai–Wuxi. This indicates that the connection between cities not only depends on the geographical proximity effect, but also on the size of the city. Generally, the larger the city, the richer its resources and the higher its network status. Therefore, if two cities are larger, the connection between them may be stronger. Moreover, the network peripheral cities such as Chizhou, Suqian, and Quzhou have relatively weak connections with core cities. In the future, it is necessary to strengthen the network status of peripheral cities, promote the integrated development of cities of different levels, and further optimize the spatial structure of urban networks.

We found that the urban network spatial structure of the Yangtze River Delta has evolved from a single-center cluster with Shanghai as its core to a multi-center network with Shanghai as its core and Nanjing, Hangzhou, and Hefei as its secondary cores. This is consistent with the actual development of the Yangtze River Delta. Some scholars have also realized that the excessive concentration of population, capital, and transportation infrastructure in large cities such as Shanghai, Hangzhou, Nanjing, and Hefei is not conducive to coordinated regional development [14]. On one hand, road–infrastructure networks are important channels for the circulation of various resources and the foundation for the development of urban networks. It is worth noting that the current road–infrastructure networks of cities in the Yangtze River Delta have strong spatial differences. For example, the road areas of cities at the edge of the network such as Chizhou, Tongling, Quzhou, and Lishui are only 20%, 18.5%, 31.4%, and 13.7% of the average in the Yangtze River Delta, respectively. On the other hand, the development of the Yangtze River Delta does not only rely on core cities such as Shanghai, Nanjing, Hangzhou, and Hefei. In the future, promoting the tilt of public resources to cities at the edge of the network and appropriately narrowing the development gap between cities will be conducive to the coordinated development of the Yangtze River Delta. In fact, since the 2009 Yangtze River Delta Cooperation and Development Conference was held, the Chinese government has begun to implement equalization development in the fields of road infrastructure, medical care, and education, and has promulgated various measures to promote the development of a multi-center network in the Yangtze River Delta. We also found that the urban network has an inverted U-shaped relationship with per capita GDP, that is, maintaining a proper GDP development gap can promote the development of urban networks, whereas too large a gap will hinder its development. Moreover, cities with similar industrial structures and government actions have stronger network connections. This shows that cities with the same cultural background and institutional environment are more inclined to connect. Finally, based on our research results, we recommend optimizing the urban network spatial structure from the following aspects:

(1) Optimize the spatial distribution of resources and guide the rational positioning of cities.
(2) Through the network spillover effect, enhance the connection strength between the network edge cities and the core cities.

(3)   Construct a multi-center network development mode to promote the integrated development of the Yangtze River Delta.

This study also has some limitations that are worth exploring further. We did not consider the function of the urban network and only focused on the evolution characteristics and patterns of urban network spatial structure, so the functions provided by cities in the network and their influence on economic and industrial development need to be further explored. Is the urban network of the Yangtze River Delta beneficial to the development of peripheral cities? Our study suggests that the density of urban networks and the degree of resource exchange between cities are gradually increasing, which are beneficial to the development of peripheral cities to a certain extent. However, some scholars worry that the urban network will negatively impact regional efficiency and fairness, and further promote the 'siphon effect' of core cities [56,57]. Therefore, these suggestions need to be thoroughly tested through a systematic study on the dynamics of the urban network. For example, we could measure changes in the centrality and intermediary centrality of node cities in the urban network, and observe the impact of the development of the urban network on cities of different levels. We could also explore the causal relationship between network centrality and the level of economic development through spatial econometric models.

**Author Contributions:** Conceptualization, Yizhen Zhang and Agus Supriyadi; Methodology, Kun Zhang and Zhi Tang; Formal analysis, Yizhen Zhang and Agus Supriyadi; Writing—original draft, Tao Wang; Writing—review and editing, Tao Wang. All authors have read and agreed to the published version of the manuscript.

**Funding:** This work is supported by the National Natural Science Foundation of China, Grant/Award Number: 41471103

**Acknowledgments:** The authors are thankful to Kun Zhang and Zhi Tang for their data preparation and visualizations.

**Conflicts of Interest:** The authors declare no conflict of interest with regard to the publication of this paper.

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
