# Peer review of "Evolution and Optimization of Urban Network Spatial Structure: A Case Study of Financial Enterprise Network in Yangtze River Delta, China"

_ijgi, doi:10.3390/ijgi9100611_

Round 1
Reviewer 1 Report
I recommend it for publication without any changes.
Author Response
Dear reviewer,
We are truly grateful to your critical comments and thoughtful suggestions on our manuscript titled "Evolution and optimization of urban network spatial structure: A case study of financial enterprise network in Yangtze River Delta,China" (Manuscript ID: ijgi-899337).
Comments and Suggestions for Authors
Point 1: I recommend it for publication without any changes.
Responses1: Thank you very much for your comments. Your comments and suggestions are very important to the scientificity and rigor of the paper. Your support and encouragement will be of great help to my future scientific research and life path. We checked the content, format, and sentences of other parts of the article to ensure that the publication requirements were met. Moreover, we also found a native speaker to polish the language of the article. Please see the certification of English editing. We sincerely thank you for your comments and wish you a happy life and smooth work.

Reviewer 2 Report
The subject has a great relevance since devising models to understand the evolution of the spatial structure between cities is important to north policies regarding urban and economic development. Models such as the presented can serve as guidelines for decision-making for a region that has been in constant growth, such as the Yangtze River Delta.
Although not a novelty in the modelling aspect – being mostly based on locational models that derive from Christaller, the paper proposes new methods for gathering spatial data using python, which combined with network analysis have very interesting results. It would be nice, however, if the authors also covered the relations of the city dimension – or their relative importance in the network – and its road-infrastructure configuration.
Methodology uses several consolidated network measures such as degree, density and modularity, all well explained and formally defined by equations.
Other measures such as Multidimensional Scaling and Quadratic Assignment Procedure are also well explained with step-by-step of models’ operation.
Other models that considered road-infrastructure networks and its configuration could also have been explored.
Language must be revised throughout all the paper, as there are many short phrases that seem disconnected from the main idea not being clear.
Major revisions are needed on introduction and conclusion.
Methodology and discussion are well written.
Overall need for restructuration of introduction and conclusion and revision of the English language.
Would be nice if in the conclusions the authors made a reflection about the road-infrastructure networks and how its configuration might be adequate or inadequate when compared to the overall city importance.
Author Response
Dear reviewer,
We are truly grateful to your critical comments and thoughtful suggestions on our manuscript titled "Evolution and optimization of urban network spatial structure: A case study of financial enterprise network in Yangtze River Delta, China" (Manuscript ID: ijgi-899337). You are very serious and professional, and put forward a lot of valuable opinions, which greatly improved the quality of the paper. Based on these comments and suggestions, we have carefully revised the original manuscript. Below you will find our point-by-point responses. Comments are shown in black, and our responses are shown in red. Moreover, you can use the "Track Changes" function in the word to view the modification traces of this article.
Comments and Suggestions for Authors
Point 1: The subject has a great relevance since devising models to understand the evolution of the spatial structure between cities is important to north policies regarding urban and economic development. Models such as the presented can serve as guidelines for decision-making for a region that has been in constant growth, such as the Yangtze River Delta.
Responses1: Thank you very much for your comments. Your comments and suggestions are very important to the scientificity and rigor of the paper. In future research, we will continue to try to combine new data and new methods to better serve the spatial governance and planning of the Yangtze River Delta and other regions of the world.
Point 2: Although not a novelty in the modelling aspect–being mostly based on locational models that derive from Christaller, the paper proposes new methods for gathering spatial data using python, which combined with network analysis have very interesting results. It would be nice, however, if the authors also covered the relations of the city dimension–or their relative importance in the network–and its road-infrastructure configuration.
Responses2: Thank you very much for your comments. Your comments and suggestions are very important to the scientificity and rigor of the paper. Based on your second and eighth suggestions, I added a discussion of the relationship between road infrastructure and urban networks in Sections 4.1, 4.3 and the conclusion of the article.
The red font in this article is the first modification, and the font with the "Track Changes" function is the second modification. The specific modifications are as follows:
The modification of “4.1 Spatial Connection Identification ” (In lines 310-320 of the paper):
In the early stages of urban network development, to obtain more externalities, cities were more inclined toward proximity and directional linkages [7]. For example, in 1990, Shanghai’s connection strength with Suzhou, Jiaxing and other surrounding cities was significantly higher than other cities. On the one hand, various forms of proximity, including geographic and cultural proximity, tend to reshape the urban network structure. On the other hand, the prerequisite for the development of urban networks is perfect road-infrastructure networks. However, road-infrastructure networks in the Yangtze River Delta were poorly developed before 1990, which restricted the cross-regional flow of resources and hindered the development of urban networks. Since 2000, Shanghai has gradually broken through administrative barriers and spatial boundaries, and has begun to connect with network edge cities such as Fuyang and Quzhou, but the connection strength remained relatively weak.
The modification of “4.1 Spatial Connection Identification ” (In lines 341-349 of the paper):
However, the network edge cities such as Chizhou, Suqian, and Quzhou have relatively weak connections with core cities. The spatial distribution of urban connections is unbalanced and showed strong network hierarchical characteristics. This is not conducive to the integration and coordinated development of the Yangtze River Delta. Moreover, related research shows that road-infrastructure networks are the prerequisite for resource circulation. Any country or city that wants to strengthen economic and industrial development and promote inter-city links must first strengthen the construction of road-infrastructure networks. In the future, it is necessary to optimize the spatial distribution of resources, strengthen the construction of road-infrastructure networks between edge cities and core cities, and improve the network status of edge cities.
Point 3: Methodology uses several consolidated network measures such as degree, density and modularity, all well explained and formally defined by equations. Other measures such as Multidimensional Scaling and Quadratic Assignment Procedure are also well explained with step-by-step of models’ operation.
Responses3: Thank you very much for your comments. Your comments and suggestions are very important to the scientificity and rigor of the paper. Your support and encouragement will be of great help to my future scientific research and life path. Moreover, we also checked the content, format, and sentences of other parts of the article to ensure that the publication requirements were met.
Point 4: Other models that considered road-infrastructure networks and its configuration could also have been explored.
Responses4: Thank you very much for your comments. Your comments and suggestions are very important to the scientificity and rigor of the paper. Although, this article studies the evolution of urban network spatial structure based on financial enterprise networks. However, we also believe that road-infrastructure networks are the basis for the development of any type of network. Therefore, based on your suggestions, we have added a discussion about the relationship between the road infrastructure network and the urban network in the article.
The red font in this article is the first modification, and the font with the "Track Changes" function is the second modification. The specific modifications are as follows:
The modification of “4.3.2 Point-axis diffusion mode ” (In lines 422-435 of the paper):
In 2000, with the further decline in the cost of intercity connection, and the rapid development of road-infrastructure networks. Core cities such as Shanghai, Hangzhou, and Nanjing rely on the Shanghai-Hangzhou and Shanghai-Nanjing expressways to gradually increase the strength of connections between cities. The strength of connections between core cities such as Shanghai, Hangzhou and Nanjing and surrounding cities have especially increased over time (Figure 7b). However, the connections between other cities remained weak. First, this indicated that the dynamic evolution of urban networks maintained notable spatiotemporal inertia. Hence, second-tier cities have more opportunities to absorb resources and economy from the top cities. Moreover, compared with small cities, second-tier cities have better economic and industrial absorption capabilities. This is not surprising as the urban network is a manifestation of the city's deep participation in the division of labor and collaboration with its advantages in resources, capital, and talent. Generally, the larger the city, the richer its resources and the higher its network status. Second, it showed that the connections between cities were generally spread along main traffic lines. In other words, urban road-infrastructure networks determined the development of urban networks to a certain extent.
The modification of “6. Conclusions” (In lines 586-611 of the paper):
We found that the urban network spatial structure of the Yangtze River Delta has evolved from a single-center cluster with Shanghai as its core to a multi-center network with Shanghai as its core and Nanjing, Hangzhou, and Hefei as its secondary cores. This is consistent with the actual development of the Yangtze River Delta. Some scholars have also realized that the excessive concentration of population, capital, and transportation infrastructure in large cities such as Shanghai, Hangzhou, Nanjing, and Hefei is not conducive to coordinated regional development [14]. On the one hand, road-infrastructure networks are important channels for the circulation of various resource and the foundation for the development of urban networks. It is worth noting that the current road-infrastructure networks of cities in the Yangtze River Delta have strong spatial differences. For example, the road areas of cities at the edge of the network such as Chizhou, Tongling, Quzhou, and Lishui are only 20%, 18.5%, 31.4%, and 13.7% of the average in the Yangtze River Delta. On the other hand, the development of the Yangtze River Delta does not only rely on core cities such as Shanghai, Nanjing, Hangzhou, and Hefei. In the future, promoting the tilt of public resources to cities at the edge of the network and appropriately narrowing the development gap between cities will be conducive to the coordinated development of the Yangtze River Delta. In fact, since the 2009 Yangtze River Delta Cooperation and Development Conference was held, the Chinese government has begun to implement equalization development in the fields of road infrastructure, medical care and education, and has promulgated various measures to promote the development of a multi-center network in the Yangtze River Delta. We also found that the urban network has an inverted U-shaped relationship with per capita GDP, that is, maintaining a proper GDP development gap can promote the development of urban networks, whereas too large a gap will hinder its development. Moreover, cities with similar industrial structures and government actions have stronger network connections. This shows that cities with the same cultural background and institutional environment are more inclined to connect. Finally, based on our research results, we recommend optimizing the urban network spatial structure from the following aspects:
Point 5: Language must be revised throughout all the paper, as there are many short phrases that seem disconnected from the main idea not being clear. Major revisions are needed on introduction and conclusion. Overall need for restructuration of introduction and conclusion and revision of the English language.
Responses5: Thank you very much for your comments. Your comments and suggestions are very important to the scientificity and rigor of the paper. According to your suggestion, the text has been checked by native speaker. Please see the certification of English editing. In addition, we have modified the text and content of the introduction and conclusions to make the logic of the article clearer.
The red font in this article is the first modification, and the font with the "Track Changes" function is the second modification. The specific modifications are as follows:
The modification of “1. Introduction” (In lines 44-97 of the paper):
Since the 1970s, production segmentation, multinational corporations, outsourcing and innovations in information technology have promoted the evolution of the spatial structure and development of urban networks [1-2]. The development of urban networks has not only accelerated the flow and reorganization of resources, but has also promoted the transition of "space of places" to "space of flows"[3]. As Taylor put it, as a new and highly efficient spatial form, the urban network plays an important role in regional governance and spatial optimization [4]. Hence, the development of urban networks reflects the evolution of the spatial structure to a certain extent, and measuring the level of urban network development is a prerequisite for spatial optimization.
In 1986, Friedman proposed the concept of a ‘world city’, which set off a wave of urban network research [5]. With the rapid development of Internet technology and modern transportation, the trend in spatial networking is becoming increasingly obvious, and increasing numbers of theories and practices related to urban networks are being introduced. Some scholars stated that the networked spatial structure enhances the externalities and spillover effects of economic development and strengthens the synergy and complementarity between cities at different levels [6-7]. Other scholars reported that the higher the degree a city is embedded in the network, the greater the opportunity for development and the lower the risk of opportunistic behavior [8-10]. With the signing of the United States-Mexico-Canada Agreement (USMCA) and the official implementation of the Horizon 2020 plan, economic and scientific cooperation between different countries and cities has been promoted, thereby further strengthening the development of urban networks [11]. Similar to some countries in North America and Europe, a series of policies to promote regional integration and spatial network development, such as Yangtze River Delta Integration and Guangdong-Hong Kong-Macao Greater Bay Area, have been implemented in China [12-14]. These phenomena reflect the importance of strengthening urban network construction as a channel for spatial optimization and high-quality urban development.
Recently, the urban network research based on the corporate network perspective has attracted increased attention from scholars [4,6,11]. As the core of the productive service industry, financial enterprises are important carriers of capital, talent, information, and services, and have gradually become a key driving force for the development of urban networks [9,15]. As revealed by Gomber [16], the study of urban networks based on financial enterprises can more accurately reflect the structure of an urban network . However, due to the difficulty and complexity of obtaining financial enterprise data, most scholars only studied static urban networks based on data in one time period; research on dynamic urban networks based on multiple time periods is lacking [17-18]. Moreover, most existing studies used the headquarters of listed financial companies and their subsidiaries to construct urban networks, but this may not accurately reflect the urban networks. Small-and medium-sized enterprises (SMEs) account for more than 99% of the total number of enterprises in the Yangtze River Delta and have become the main force driving economic development(Retrieved from https://www.qcc.com/ in July 2019). If we focus too on the urban network formed by listed financial companies, it may lead to too much research focusing on large cities, causing small cities to be neglected [19].
In order to fill the above gaps, we provide the following improvements. First, we collected data on the headquarters and branches of all banks, securities and insurance companies in the Yangtze River Delta region from 1990 to 2017 using Python programming language. Secondly, combined with geographic information system(GIS) spatial analysis, social network analysis and multidimensional scale models, we explored the evolution of the urban network spatial structure in the Yangtze River Delta and propose corresponding optimization measures. The main advantage of using this data is that financial companies are not only important carriers for the allocation of resources such as capital and technology, but also sensitive to market changes [20]. Moreover, the corporate network formed by the cross-regional layout of financial firms has become a key driving force for shaping urban networks [15]. Finally, two questions needed to be answered: What are the characteristics of the evolution of urban network spatial structure and its driving mechanism, and what is the evolution mode of urban networks in the Yangtze River Delta?
The paper is structured as follows: First, we briefly review the relevant literature in the field of urban networks. Second, we introduce the research areas, data processing and methods. Third, we combine GIS spatial analysis, a modularity model, and a multidimensional scale model to explore the evolution characteristics of urban network spatial structure. We also discuss the factors influencing the urban network and provide some suggestions for the optimization of the urban network spatial structure. Finally, we summarize our conclusions and future research directions.
The modification of “6. Conclusions” (In lines 565-630 of the paper):
The urban network is an important method of spatial optimization, and measuring the development level of the urban network is a prerequisite for spatial optimization [24]. In this study, we first used Python programming language to crawl the data of financial companies of different sizes and levels in the Yangtze River Delta in 1990, 2000, 2010, and 2017. Secondly, combined with the modularity and MDS models, we explored the network community division and network development model respectively. Finally, we explored the factors influencing the urban network and provided some suggestions for the optimization of the urban network spatial structure. Our findings enrich and deepen the understanding of the urban network characteristics of the Yangtze River Delta, and provide a reference for optimizing urban network spatial structure.
Consistent with other research, we found that different forms of proximity, including cultural and geographic proximity, tend to shape the urban network structure [6,40,42]. We also found that the network connection has a certain path dependence effect due to the embeddedness of the urban network. For example, a lock-in effect was found in the inter-city connections in the Yangtze River Delta, which was mainly concentrated between Shanghai-Hangzhou, Shanghai-Nanjing, Shanghai-Suzhou and Shanghai-Wuxi. This indicates that the connection between cities not only depends on the geographical proximity effect but also on the size of the city. Generally, the larger the city, the richer its resources and the higher its network status. Therefore, if two cities are larger, the connection between them may be stronger. Moreover, the network peripheral cities such as Chizhou, Suqian, and Quzhou have relatively weak connections with core cities. In the future, it is necessary to strengthen the network status of peripheral cities, promote the integrated development of cities of different levels, and further optimize the spatial structure of urban networks.
We found that the urban network spatial structure of the Yangtze River Delta has evolved from a single-center cluster with Shanghai as its core to a multi-center network with Shanghai as its core and Nanjing, Hangzhou, and Hefei as its secondary cores. This is consistent with the actual development of the Yangtze River Delta. Some scholars have also realized that the excessive concentration of population, capital, and transportation infrastructure in large cities such as Shanghai, Hangzhou, Nanjing, and Hefei is not conducive to coordinated regional development [14]. On the one hand, road-infrastructure networks are important channels for the circulation of various resource and the foundation for the development of urban networks. It is worth noting that the current road-infrastructure networks of cities in the Yangtze River Delta have strong spatial differences. For example, the road areas of cities at the edge of the network such as Chizhou, Tongling, Quzhou, and Lishui are only 20%, 18.5%, 31.4%, and 13.7% of the average in the Yangtze River Delta. On the other hand, the development of the Yangtze River Delta does not only rely on core cities such as Shanghai, Nanjing, Hangzhou, and Hefei. In the future, promoting the tilt of public resources to cities at the edge of the network and appropriately narrowing the development gap between cities will be conducive to the coordinated development of the Yangtze River Delta. In fact, since the 2009 Yangtze River Delta Cooperation and Development Conference was held, the Chinese government has begun to implement equalization development in the fields of road infrastructure, medical care and education, and has promulgated various measures to promote the development of a multi-center network in the Yangtze River Delta. We also found that the urban network has an inverted U-shaped relationship with per capita GDP, that is, maintaining a proper GDP development gap can promote the development of urban networks, whereas too large a gap will hinder its development. Moreover, cities with similar industrial structures and government actions have stronger network connections. This shows that cities with the same cultural background and institutional environment are more inclined to connect. Finally, based on our research results, we recommend optimizing the urban network spatial structure from the following aspects:
- Optimize the spatial distribution of resources and guide the rational positioning of cities.
- Through the network spillover effect, enhance the connection strength between the network edge cities and the core cities.
- Construct a multi-center network development mode to promote the integrated development of the Yangtze River Delta.
This study also has some limitations that are worth exploring further. We did not consider the function of the urban network? We only focused on the evolution characteristics and patterns of urban network spatial structure, so the functions provided by cities in the network and their influence on economic and industrial development need to be further explored. Is the urban network of the Yangtze River Delta beneficial to the development of peripheral cities? Our study suggests that the density of urban networks and the degree of resource exchange between cities are gradually increasing, which are beneficial to the development of peripheral cities to a certain extent. However, some scholars worry that the urban network will negatively impact regional efficiency and fairness, and further promote the ‘siphon effect’ of core cities [56-57]. Therefore, these suggestions need to be thoroughly tested through a systematic study on the dynamics of the urban network. For example, we could measure changes in the centrality and intermediary centrality of node cities in the urban network, and observe the impact of the development of the urban network on cities of different levels. We could also explore the causal relationship between network centrality and the level of economic development through spatial econometric models.
Point 6: Methodology and discussion are well written.
Responses6: Thank you very much for your comments. Your comments and suggestions are very important to the scientificity and rigor of the paper. Your support and encouragement will be of great help to my future scientific research and life path. Moreover, we also checked the content, format, and sentences of other parts of the article to ensure that the publication requirements were met.
Point 7: Would be nice if in the conclusions the authors made a reflection about the road-infrastructure networks and how its configuration might be adequate or inadequate when compared to the overall city importance.
Responses7: Thank you very much for your comments. Your comments and suggestions are very important to the scientificity and rigor of the paper. There is an old saying in China, "If you want to be rich, you must first build roads." This shows the importance of road infrastructure to the development of economy, industry and urban networks. According to your 2nd and 8th suggestions, I added the relationship between road infrastructure and urban network in the third section and conclusion of the article.
The red font in this article is the first modification, and the font with the "Track Changes" function is the second modification. The specific modifications are as follows:
The modification of “6. Conclusions” (In lines 615-692 of the paper):
The urban network is an important method of spatial optimization, and measuring the development level of the urban network is a prerequisite for spatial optimization [24]. In this study, we first used Python programming language to crawl the data of financial companies of different sizes and levels in the Yangtze River Delta in 1990, 2000, 2010, and 2017. Secondly, combined with the modularity and MDS models, we explored the network community division and network development model respectively. Finally, we explored the factors influencing the urban network and provided some suggestions for the optimization of the urban network spatial structure. Our findings enrich and deepen the understanding of the urban network characteristics of the Yangtze River Delta, and provide a reference for optimizing urban network spatial structure.
Consistent with other research, we found that different forms of proximity, including cultural and geographic proximity, tend to shape the urban network structure [6,40,42]. We also found that the network connection has a certain path dependence effect due to the embeddedness of the urban network. For example, a lock-in effect was found in the inter-city connections in the Yangtze River Delta, which was mainly concentrated between Shanghai-Hangzhou, Shanghai-Nanjing, Shanghai-Suzhou and Shanghai-Wuxi. This indicates that the connection between cities not only depends on the geographical proximity effect but also on the size of the city. Generally, the larger the city, the richer its resources and the higher its network status. Therefore, if two cities are larger, the connection between them may be stronger. Moreover, the network peripheral cities such as Chizhou, Suqian, and Quzhou have relatively weak connections with core cities. In the future, it is necessary to strengthen the network status of peripheral cities, promote the integrated development of cities of different levels, and further optimize the spatial structure of urban networks.
We found that the urban network spatial structure of the Yangtze River Delta has evolved from a single-center cluster with Shanghai as its core to a multi-center network with Shanghai as its core and Nanjing, Hangzhou, and Hefei as its secondary cores. This is consistent with the actual development of the Yangtze River Delta. Some scholars have also realized that the excessive concentration of population, capital, and transportation infrastructure in large cities such as Shanghai, Hangzhou, Nanjing, and Hefei is not conducive to coordinated regional development. [14]. On the one hand, road-infrastructure networks are important channels for the circulation of various resource and the foundation for the development of urban networks. It is worth noting that the current road-infrastructure networks of cities in the Yangtze River Delta have strong spatial differences. For example, the road areas of cities at the edge of the network such as Chizhou, Tongling, Quzhou, and Lishui are only 20%, 18.5%, 31.4%, and 13.7% of the average in the Yangtze River Delta. On the other hand, the development of the Yangtze River Delta does not only rely on core cities such as Shanghai, Nanjing, Hangzhou, and Hefei. In the future, promoting the tilt of public resources to cities at the edge of the network and appropriately narrowing the development gap between cities will be conducive to the coordinated development of the Yangtze River Delta. In fact, since the 2009 Yangtze River Delta Cooperation and Development Conference was held,the Chinese government has begun to implement equalization development in the fields of road infrastructure, medical care and education, and has promulgated various measures to promote the development of a multi-center network in the Yangtze River Delta. We also found that the urban network has an inverted U-shaped relationship with per capita GDP, that is, maintaining a proper GDP development gap can promote the development of urban networks, whereas too large a gap will hinder its development. Moreover, cities with similar industrial structures and government actions have stronger network connections. This shows that cities with the same cultural background and institutional environment are more inclined to connect. Finally, based on our research results, we recommend optimizing the urban network spatial structure from the following aspects:
- Optimize the spatial distribution of resources and guide the rational positioning of cities.
- Through the network spillover effect, enhance the connection strength between the network edge cities and the core cities.
- Construct a multi-center network development mode to promote the integrated development of the Yangtze River Delta.
This study also has some limitations that are worth exploring further. We did not consider the function of the urban network? We only focused on the evolution characteristics and patterns of urban network spatial structure, so the functions provided by cities in the network and their influence on economic and industrial development need to be further explored. Is the urban network of the Yangtze River Delta beneficial to the development of peripheral cities? Our study suggests that the density of urban networks and the degree of resource exchange between cities are gradually increasing, which are beneficial to the development of peripheral cities to a certain extent. However, some scholars worry that the urban network will negatively impact regional efficiency and fairness, and further promote the ‘siphon effect’ of core cities [56-57]. Therefore, these suggestions need to be thoroughly tested through a systematic study on the dynamics of the urban network. For example, we could measure changes in the centrality and intermediary centrality of node cities in the urban network, and observe the impact of the development of the urban network on cities of different levels. We could also explore the causal relationship between network centrality and the level of economic development through spatial econometric models.

This manuscript is a resubmission of an earlier submission. The following is a list of the peer review reports and author responses from that submission.
Round 1
Reviewer 1 Report
The paper appears to set out at assembling a representation of the urban network in the Yangtze River Delta based on co-occurrence of offices of financial enterprises in four temporal snapshots between 1990-2017. Through this, the authors aim at identifying and characterizing the model of the regional urban network's evolution. The manuscript in general does overall merit publication. However, it suffers from confused communication of methods and unclear description of the data and sequence in which the methodology has been deployed making it difficult to truly examine the manuscript's rigor. I'd recommend major revision with authors encouraged to address the following points.
Major comments
- While I find the manuscript of interest, the authors have not made a case as to why their contribution is a match for journal to which they have submitted their work as the spatial and/or data mining aspect of the work are only reviewed in passing.
- Throughout the manuscript, the authors repeatedly make references to the "optimization" of the urban network. These are, however, never accompanied with a description or elaboration as to what is being optimized and what is the objective function of the optimization. This is a critical oversight as the optimization angle appears a core point for the authors in terms of the practical use of their work.
- The manuscript is unclear as to what data is collected using "Python programming language". The authors provide a cursory count of institutions that have supposedly been collected for each time interval but it is unclear where the number represent total unique enterprises each with various branches across the 41 cities or whether they are the total number of offices for an unspecified number of unique enterprises. Ideally, if this data is publicly available for the authors to have crawled it, they should provide a curated data set to enable reproduction of their study.
- In equation 1, the authors attempt to describe the fashion in which edges and their weights have been calculated for each urban dyad. The equation is missing index for companies, k, used in the sum operator and for cities, i and j, and hence is overall unclear as to how the values of each C and R are calculated. What values can these Cs take? Is Cch, for example, 1 if a company has headquarters in both city i and j? These need to be much better explained and demonstrated not least for reproducibility of the study.
- In their description of the modularity, the authors make references to two articles which appear to be only available in Chinese which does not facilitate contextualization of the method and the equation which can be found in Newman (2004) with an implementation algorithm for it available in Blondel et al. (2008). Most importantly, however, is that it is unclear whether the authors are using the R values from the previous section as weights for these modularity estimations.
- The authors then half describe a multidimensional scaling procedure but remains absolutely unclear as to what data has been used in the procedure and how it actually affects and integrates with the rest of the manuscript. References to "transposed vector space" and "coordinate matrix" within Figure 6 appear without any elaboration and or explanation. Despite a reference to the MDS determining the evolution mode in the conclusion, it is not clear what part of the results in fact correspond to the method.
- Similar to the modularity section, the sole reference for the Quadratic Assignment Procedure appears to be only available in Chinese. This requires further elaboration as to how the method works and/or use of more accessible and relevant references.
- Due to the previous points, it is difficult to judge what the edges and their weights shown in Figures 7 and 8 in fact represents. Due these represent the R values calculated previously, if so it would be helpful for the authors to include figure showing edge weight and degree distribution plots either in the main text and/or as supplementary material.
- For the evolution modes presented in Figure 9, the authors argument on point-axis diffusion appears to rest on a spatial basis, ie, the existence of a point-axis structure in the geospatial development of the delta. There is, however, no evidence for this as the figure obviously does not show spatially embedded networks, unlike Figure 7, and the only difference between the panels b and c seems to be layout in which the authors have chosen to visualize the network. I am inclined to think that based on the conclusion these are rather based on the MDS analysis, however, actual description of what the authors may have done in the figure is absent.
- Lastly, the authors describe their QAP regression model. Overall, I am unclear as to what the dependent variable Y in fact is. Guessing based on the application of the QAP in social networks, I am assuming the authors may have used the R values? On more specific unresolved questions, the parameter Λ in the equation is left undescribed. The authors also describe having used both GDP per head and its squared value as independent variables. I presume that the authors have used the difference between the GDP per head of each city pair as an independent variable and hence would have had a mix of negative and positive values. While this may then explain why they have used squared values for the observation of the U-shaped response, it does not explain using both the difference and its squared values simultaneously as this would introduce correlated independent variables into the model. It also requires better justification as to why the authors have opted for the squared values rather than simply using the absolute difference of the GDP per head.
Minor comments
- References to "cyberspace" on p2l43 appear unrelated to the point the authors seem to making in lines before and after!
- On p2l50-53, the authors appear to be contrasting two ideas that are not contradictory. The referee is not sure whether this a made choice of language or the the ideas are meant to be contradictory and contrasting.
- On p2l55, it is unclear how the USMCA strengthens the development of urban networks.
- On p2l68, the authors could do with spelling out what is "dynamic correlation data".
- On p4l151, land area is quoted as a percentage point, is this relative to the over all land mass of continental China?
- On p6 Equations 1 and 2 are missing indexes or have mismatching indexes between the equation and explanatory text afterwards.
- On p11 Figure 8 requires labeling for years.
References
Vincent D Blondel et al J. Stat. Mech. (2008) P10008
M E J Newman Phys. Rev. E 70 (2004) 056131
Reviewer 2 Report
The article seems to be very interesting. I have some technical comments:
Figure 4. the name Shanghai is in lowercase. There are no marked provincial
borders.
Although this is not the subject of the research, the text lacks a table with the
characteristics of the analyzed values and their change for the analyzed units
(e.g. mean values). In the text there is only a chart for the number of financial
companies in the Yangtze River Delta. He can add such a table to the text as an attachment.
Reviewer 3 Report
The paper is very interesting but in my opinion it needs to be improved both in its structure and in its conclusions.
The correspondence with the figures is not always respected and this makes difficulties for the reader.
In its present form the paper cannot be published, according to my opinion
My suggestions are in the file attached
